# Digital Twin Haptic Robotic Arms: Towards Handshakes in the Metaverse

Mohd Faisal [1], Fedwa Laamarti [1,2] and Abdulmotaleb El Saddik [1,2,*]

1. School of Electrical Engineering and Computer Science, University of Ottawa, Ottawa, ON K1N 6N5, Canada; mmohd055@uottawa.ca (M.F.); flaam077@uottawa.ca (F.L.)
2. Department of Computer Vision, Mohamed bin Zayed University of Artificial Intelligence, Masdar City, Abu Dhabi P.O. Box 5224, United Arab Emirates
* Correspondence: elsaddik@uottawa.ca

**Abstract:** More daily interactions are happening in the digital world of the metaverse. Providing individuals with means to perform a handshake during these interactions can enhance the overall user experience. In this paper, we put forward the design and implementation of two right-handed underactuated Digital Twin robotic arms to mediate the physical handshake interaction between two individuals. This allows them to perform a handshake while they are in separate locations. The experimental findings are very promising as our evaluation shows that the participants were highly interested in using our system to shake hands with their loved ones when they are physically separated. With this Digital Twin robotic arms system, we also found a correlation between the handshake characteristics and personality traits of the participants from the handshake data collected during the experiment.

**Keywords:** digital twin; metaverse; haptics; anthropomorphic robotic hand; robotic arm; human–robot interaction (HRI); handshake; personality





## 1. Introduction

The development of digital immersive technology was accelerated with the introduction of the 'Metaverse', the future of the internet. This technology is expected to reduce the difference between the physical and the virtual worlds by providing a fully immersive experience to the users of the internet [1]. Humans are social beings, and physical touch plays an important role in our well-being. Conversely, many interactions that used to routinely happen in the real world now take place in the virtual world. The existing research on interactions between the virtual world and real world may provide some answers as to why this shift has occurred.

The concept of a Digital Twin (DT) [2] was introduced by Michael Grieves. According to him, a DT is actionable, which means that the model can simulate the effects of external forces applied to elements in the model itself. Grieves's definition was extended in [3] to include Digital Twins of humans and living things in general. The work in [4] introduced a DT architecture by establishing the existing relations between the different Real Twins (RTs) and sending their data to the cloud. The DT suggested in [4] can model relations and interact directly with the real world by producing a depiction of a Real Twin. The work in [5] introduced an ecosystem to implement DTs as well as presented all the necessary tools and technologies to implement a DT for health and well-being [6].

For a DT to provide an immersive experience in the form of interaction between individuals, a 'physical body' needs to be provided. This physical representation of the Digital Twin, referred to as a Robo Twin in our previous work [7], should have a body that feels as close as possible to the body of a human and the capacity to capture touch-related information, as in case of a human interaction. The fields of haptics and robotics are beginning to converge around the idea of touch [8]. In recent years, human–robot

interactions (HRIs) have gained increasing attention from the research community as more and more robots are developed and introduced in the context of providing social assistance to humans. One such natural action that is frequently employed in various social situations between two people is shaking hands. Handshakes are typically the initial non-verbal exchange in a social setting, helping to establish the tone and shape first impressions [9]. As a result, it is an extremely important social behavior that people engage in, and thus robots should be able to replicate this act. However, few studies have been conducted on human–robot handshaking [10]. Therefore, in this paper, we focus on the handshake aspect of human–robot interaction.

The rest of the paper is organized as follows. The next section presents the previous works on modeling a human–robot handshake and designs of anthropomorphic robotic hands inspired by human hand anatomy. Section 3 describes the methodology and the architecture of the proposed system followed by its design and implementation in Sections 4 and 5, respectively. Section 6 details the handshake experiment and presents the qualitative and quantitative analysis of the overall system. Limitations of this study are highlighted in Section 7 and finally, concluding remarks and suggestions for future works are provided in Section 8.

## 2. Related Work

There is a scarcity of literature on this topic. To the best of our knowledge, there is currently no robotic arm capable of mimicking a handshake between two physically distant humans. The background work, therefore, is categorized into two parts: human–human handshake analysis and robotic designs able to implement human–robot handshakes inspired by human hand anatomy.

Human–human handshake analysis: In [9], a conceptual framework for handshaking between humans and robots is presented, which divides the handshake into three stages: hand reaching, hand gripping and shaking and synchronization, emphasizing the fact that each part of the movement at various stages needs to be more human-like for a more realistic perception. Another study [11] utilizes wavelet transformation to thoroughly evaluate the physical properties of a handshake, such as the acceleration of the human arm. It was found that the handshaking phases in [9] can be described qualitatively using wavelet scalogram analysis. The study conducted in [12] measures the quantitative aspect of the human handshake, such as the duration, grip strength and rate of rhythmic motion, whereas a design of a wearable haptic measurement glove (HMG) is presented in [13], wherein inertial and force data for handshakes were gathered to create an effective control algorithm for realistic human–robot interaction. A bio-inspired controller design based on the CPG model is implemented in [14] to imitate a more realistic behavior of robotic arms during a handshake. The inverse kinematic approach is used for Sophia-Hubo's right arm [15] for extending the arm towards a person and for controlling the torque while shaking hands. Comparing the existing models for physical human–robot interaction (pHRI), a detailed guide on how a more realistic handshake between humanoid robots and their human companions can be achieved is presented in [16].

Robotic designs to implement a human–robot handshake: Based on existing human–robot handshake analysis and inspired by human hand anatomy, some authors have designed and implemented robotic end-effectors similar to a human hand to realistically produce a human–robot handshake. However, these designs were not close to a human-like body as they used existing robotic arms such as Katana [14], KUKA LWR [17], KUKA LWR 4+ [18], Meka [19] and Pisa/IIT, giving a more mechanical feel and lowering the overall acceptance of HRI. They closely reflect some of the characteristics of human handshakes, such as human palm compliance [17,18], finger grasping [18] and the closed-loop hand control [20] exhibited by humans while performing a handshake.

Humanoid robots are very complex anthropomorphic machines that resemble humans and possess the ability to use the same environment as humans [21]. Since having human-like bodily movements is crucial to HRI's adoption [9], existing human-like robots,

such as ASIMO [22,23], HRP-1 [24], HRP-2 [25], HRP-3 [26], HRP-4 [27], HOAP-1 [28], NAO Robot [29,30] and HBS-1 [31], can be used as a building platform for the physical representation of a Digital Twin. However, these advanced humanoid systems are highly priced and more complex. Incorporating additive manufacturing technology in the design and fabrication of humanoid robots is a promising solution to reduce their cost [7]. The authors of [17,32–36] put forward a customized 3D-printed design of an anthropomorphic robotic hand with various underactuated design techniques for various applications. The Inmoov design [37] with its underactuation technique was used by the authors of [38,39] for robotic applications involving human interaction because of its open-source design, low cost and human-like appearance. Since most of the available hand designs are restricted to one specific sensing functionality, the open-source Inmoov design seems to be the best available option for a robotic hand, as the design can be easily customized. Moreover, this design is one of the best examples of an economical humanoid robot [7] whose parts can be easily 3D-printed and assembled for a robust design.

## 3. Methodology

The main purpose of the proposed concept in this paper is to allow two individuals, say Real Twin A and Real Twin B, to perform a handshake, as illustrated in Figure 1. Robo Twin A and Robo Twin B will sense the handshake data from Real Twin A and Real Twin B, respectively, and then simultaneously exchange the sensed data over a communication network. It is very interesting to note that when the Robo Twin is in its actuation state, it will act as a virtual copy of the other connected Robo Twin, representing the physical state of the Real Twin on the other side. To summarize, Robo Twin A goes in the actuation state, providing haptic feedback based on the sensed data received from Robo Twin B over the communication network, thereby acting as a virtual copy of Robo Twin B so that the Real Twin A feels the physical state of Real Twin B and vice versa.

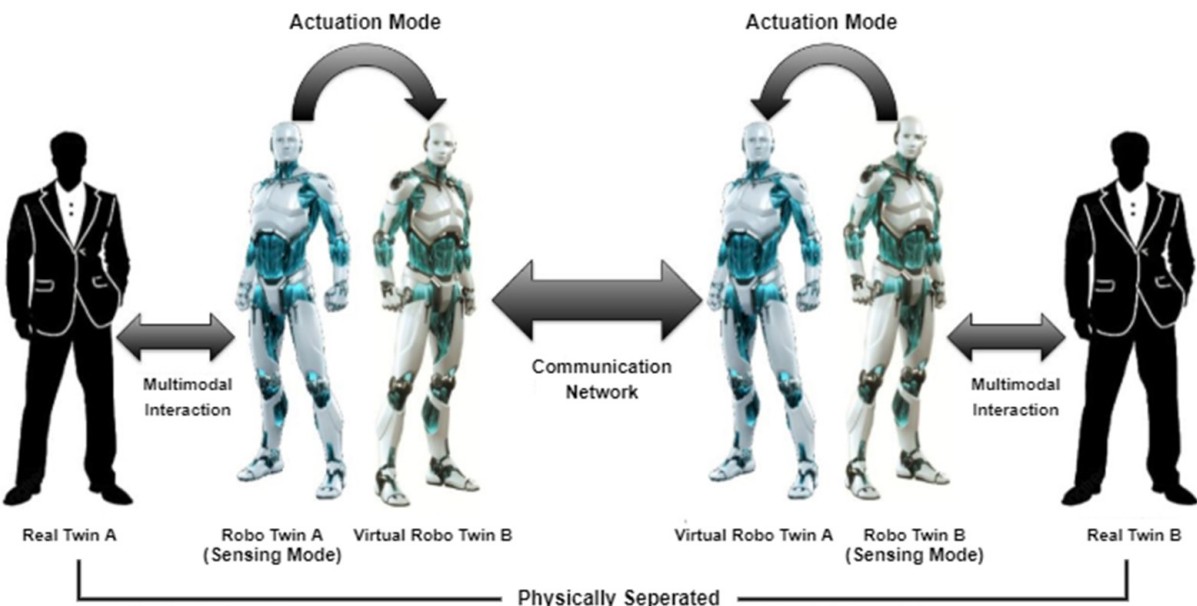

**Figure 1.** The proposed concept of Digital Twin allows physically separated individuals to perform a handshake.

The architecture of our system is given in Figure 2. It relies on sensing, processing, data collecting, data transmitting and receiving modules to achieve the task of handshaking from a distance. The task performed by each of these modules is briefly described below:

- **Sensing Module:** This module is activated as soon as the system is powered on. The main duty performed by this module is sensing the raw sensory data from the Real Twin while in contact with the Robo Twin hand, performing a handshake. All the raw

signals received by the sensors are then sent to the processing module, where they are converted to their corresponding physical values by the controller.

- **Processing Module:** This module performs the processing of the received raw analog signals from the sensors by converting them into meaningful interpreted physical data. It also actuates the actuators upon receiving the processed data from the other Robo Twin's processing module. Additionally, more importantly, upon receiving any nonzero signal from the force sensors located on the Robo Twin's hand, it then generates the control signals for the inward grasp of the subject's hand. These control signals actuate the servo motors of each finger of the Robo Twin's hand to grasp the subject's hand for a handshake. When the subject withdraws contact from the located force sensors on the Robo Twin's hand, it then generates the control signals for the outward grasp motion of the Robo Twin's fingers, thereby releasing the user's hand and ending the handshake. While performing all the above-mentioned tasks, this processing module simultaneously sends the processed sensory data to the data-collecting and data-transmitting and -receiving modules for real-time recording and transmission.
- **Data-Collecting Module:** This part of the system's architecture is responsible for the real-time recording of the sensory data while Robo Twin is performing a handshake with the user. This module is triggered by the output of the processing module, i.e., as soon as the interpreted physical data are made available.
- **Data-Transmitting and -Receiving Module:** This module establishes communication between the two Robo Twin arms for simultaneous and bidirectional transfer of sensory data.

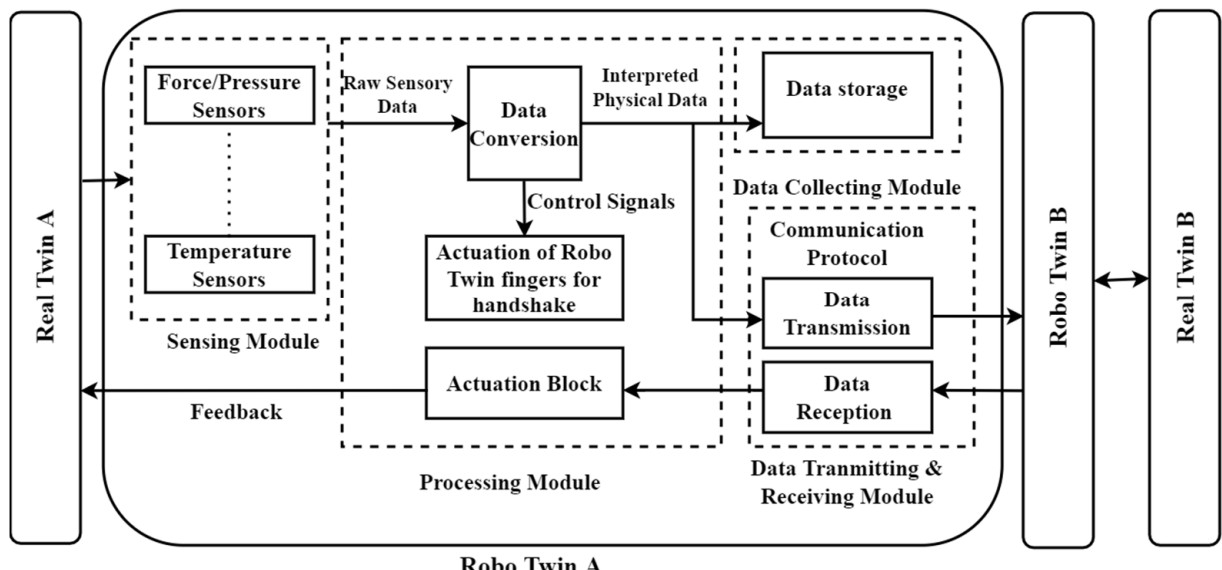

**Figure 2.** Proposed system's architecture.

## 4. Design of Robo Twin Arm

### 4.1. Robo Twin's Arm Structure

We decided to go with the Inmoov robot arm design for our Robo Twin's arm structure. All the parts of the Robo Twin arms were 3D-printed on an Ultimaker2+ printer using FDM (Fused Deposition Modeling) technology and PLA (Polylactic Acid) as the printing material. The arm structure consists of three parts:shoulder, bicep and forearm and hand. The shoulder and bicep parts were printed with an infill of 70%, a wall thickness of 2.5 mm, a nozzle size of 0.6 mm and without support. The forearm and hand parts used the same settings except for an infill of 80% and a wall thickness of 2 mm. The gears and fingers were printed with the highest quality possible with an infill of 100%, a wall thickness of 3 mm, a nozzle size of 0.4 mm and without support and raft settings. The fingers size are almost the

same size as that of an adult male human, while the forearm was made a little longer and wider to accommodate the standard-size servo motors for fingers and wrist actuation.

Although the actuation of the bicep and shoulder in the Robo Twin arm is important for creating natural and realistic handshakes, we would like to point out that in our present research, the focus was mostly on the actuation of the hand and forearm components. However, in a future phase of this study, the actuation of these components will be achieved by housing two HS-805BB heavy-duty servo motors in each, with a maximum torque capacity of up to 10 kg, providing each part with two degrees of freedom.

A highly underactuated mechanism is used to control the motion of the Robo Twin hand specifically the distal, middle and proximal phalanxes of each finger, using only a single actuator. The actuator used for the motion of these phalanxes is an MG996R servo motor placed inside the forearm. The plastic servo horns that came with the servo motors were replaced with 3D-printed servo pulleys through which a fine braided fishing line was passed, as shown in Figure 3.

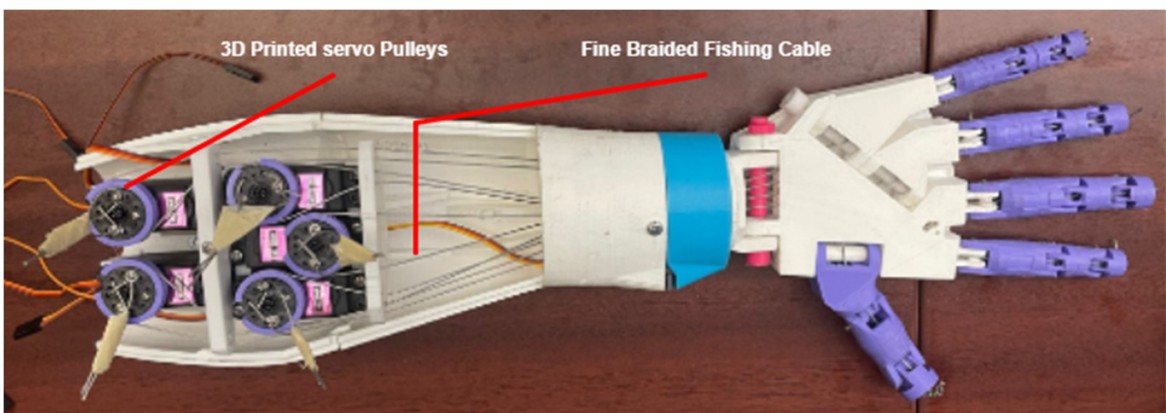

**Figure 3.** Forearm and hand of a Robo Twin with housed servo motors for finger actuation via fine braided fishing line.

This braided fishing line makes its way to the fingertips through holes integrated within the design, acting just like the tendons in the human body. At the servo end, the string is wound around the servo pulley and tied at one of its edges so that when the servo is operated, it rotates, and the string is pulled further to its edge, causing the fingertip to move towards the palm. Therefore, the tension in the string causes the distal phalanx to bend inwards. As the tension keeps increasing due to the torque provided by the servo motor, the middle phalanx also curls in, followed by the proximal phalanx. Hence, the curling motion of the finger, which constitutes 3 degrees of freedom, is achieved only one actuator. The fingers can be returned to their normal resting position by rotating the servos in the opposite direction at the same angle.

### 4.2. Sensor and Actuator Placement

As mentioned in [7], there is a lack of research on sensor and actuator placement on robotic hands, especially for a human–robot handshake application. Due to this scarcity of research, we conducted an elementary and straightforward experiment to determine the best locations for the sensor and actuator points on our Robo Twin hand for its optimal performance in performing a handshake experience as close to a real one as possible.

As a part of this experiment two human male subjects performed a handshake, say Person A and Person B, while both wearing a white silicone hand glove. In the first step of this experiment, Person A played the role of Robo Twin and Person B played the role of the Real Twin, as represented in Figure 4a. The glove of Person B was painted black, as shown in Figure 4c, to record the imprints of Person B's hand. Both subjects were asked to perform a firm handshake, so that the regions of contact were imprinted on the glove during the handshake. After the handshake was complete, the glove of Person A was analyzed to

determine the areas of the hand that came into contact and experienced the most force applied by Person B. In the second step, the same procedure was repeated, but this time the roles of Person A and Person B were swapped; Person B then played the role of a Robo Twin and Person A played the role of a Real Twin, as represented in Figure 4b.

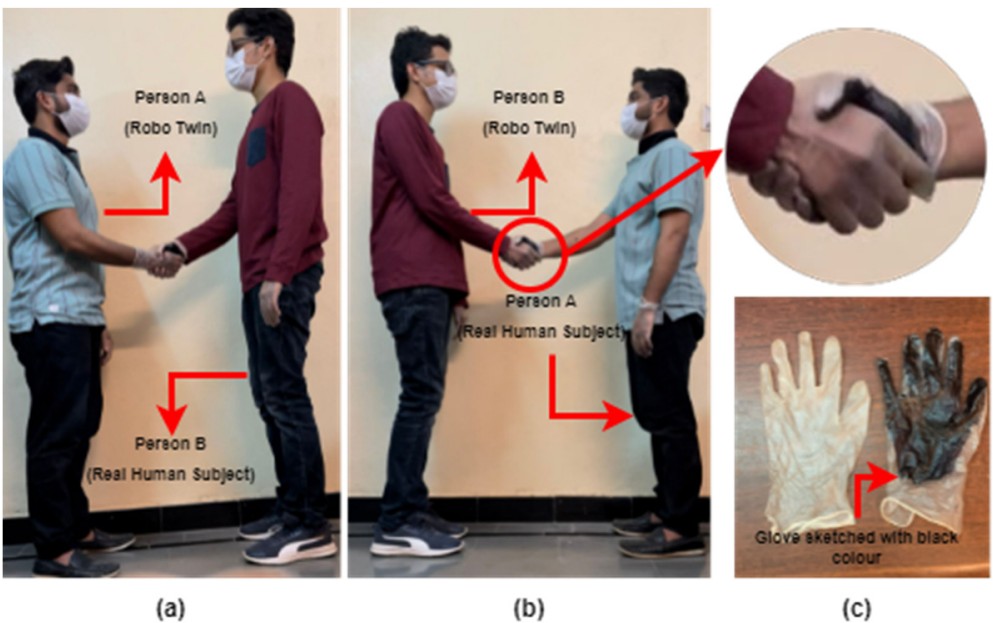

**Figure 4.** Handshake experiment to locate optimal sensor and actuator points on Robo Twin's hand for handshaking purpose. (**a**) Handshake between Person A as Robo Twin and Person B as a Real Human subject; (**b**) Handshake between Person B as Robo Twin and Person A as a Real Human Subject; (**c**) Right hand glove painted with black color for Person B.

Finally, the imprints on the gloves from both stages of the experiment were examined, and the overlapping regions of the right hand were drawn to locate the optimal position of sensors and actuators.

The regions R1, R2 and R3 highlighted in red in Figure 5a were found to have the most contact with the Real Twin's hand during a handshake. Region R1 is the area where the force applied by the index, middle, ring and pinky fingers of the real subject were detected, and similarly regions R2 and R3 are the areas where the force applied by the Real Twin's thumb was detected. Therefore, the discussed regions can be equipped with sensors to measure the force applied by the Real Twin's hand while performing a handshake.

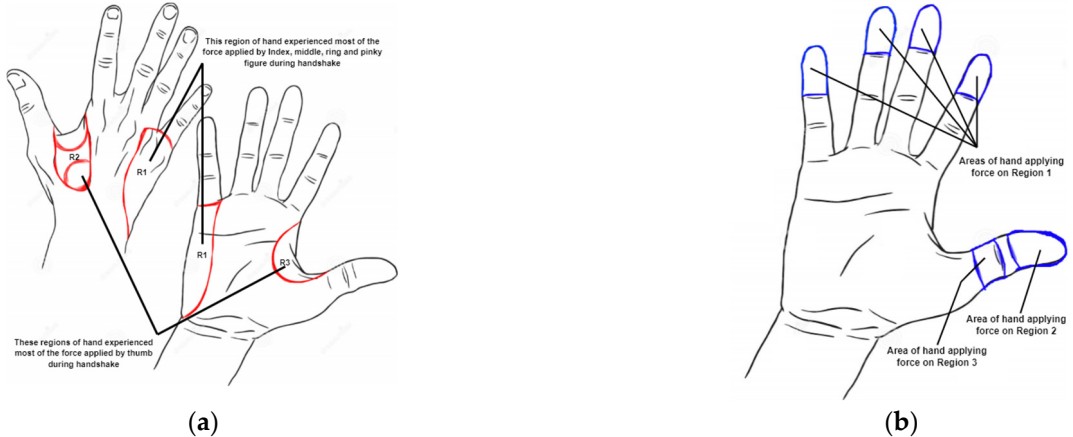

(**a**)       (**b**)

**Figure 5.** (**a**) Mapped regions on the right hand for sensor placement; (**b**) regions of the hand that applied the most force during the handshake.

The regions highlighted in blue in Figure 5b, R1, R2 and R3, are the areas of the Real Twin's hand, that applied the most force on the Robo Twin's hand. Therefore, force actuators can be placed in these areas on the Robo Twin's hand to generate the equivalent haptic feedback, while performing a handshake.

To effectively locate the temperature sensor points, we carefully analyzed the above experimental results for the regions of hand coming into contact with one another while performing a handshake and the parts of the hands responsible for sensing the handshake. These findings are illustrated in Figure 6.

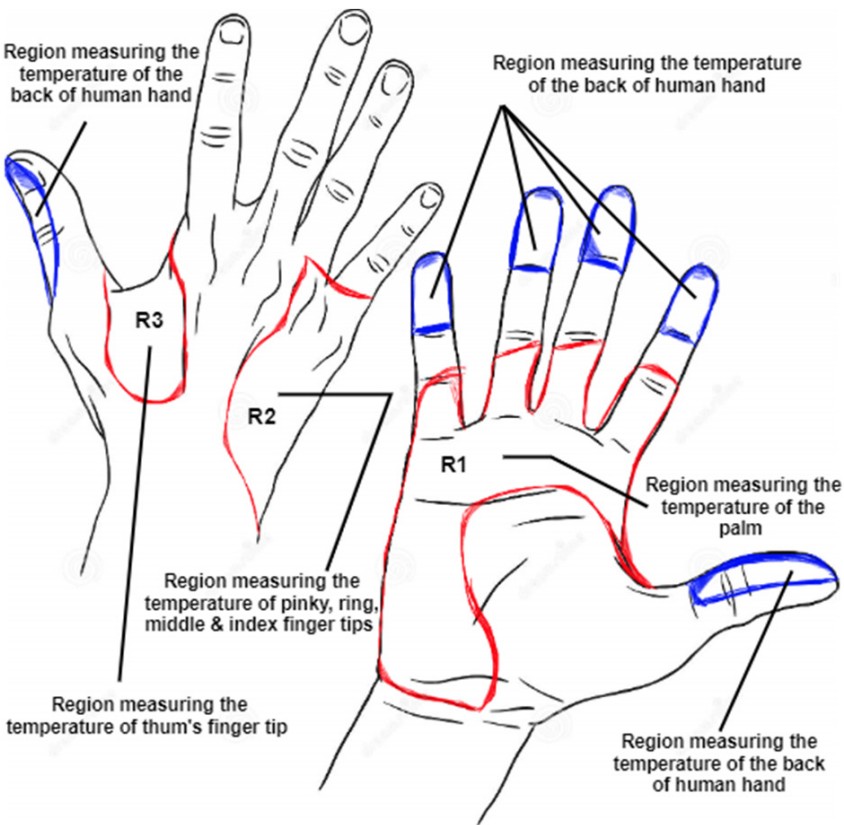

**Figure 6.** Measuring the inside and outside temperature of different regions of the hand.

Regions R1, R2 and R3 highlighted in red in Figure 6 are those that sense the inside temperature of the Real Twin's hand, while the areas highlighted in blue sense the temperature of the back side of the Real Twin's hand during a handshake. Region R1 can be used to measure the temperature of the Real Twin's palm and regions R2 and R3 can be used to sense the temperature of the Real Twin's ring, middle, index and pinky finger and thumb, respectively.

## 5. Implementation

### 5.1. Force Sensors

Sensing force is one of the crucial tasks of our Robo Twin, which was designed and developed for the purpose of performing a handshake with real human subjects. The work in [7] indicates that the force-sensing resistor (FSR) is the most widely used technology for determining the contact forces in applications involving a robotic hand. This is not only because of its low cost and ease of implementation, but also its relatively high sensitivity compared to capacitive tactile sensors and piezoelectric polymer films.

Therefore, we incorporated two different force-sensing resistors (FSRs) in our Robo Twin's hand: the Sparksfun FSR 402 and the Tekscan FlexiForce A401. The Sparksfun FSR 402, manufactured by Interlink Electronics, features a 0.5″ sensing diameter and

an actuation force as low as 0.1 N. It has a sensitivity range of up to 100 N, providing continuous resolution. The operating temperature range for this sensor is between $-30\ ^{\circ}$C and $+70\ ^{\circ}$C. On the other hand, the Tekscan FlexiForce A401 has a 1″ sensing diameter and can measure forces up to 111 N. It operates within a temperature range of $-40\ ^{\circ}$C to $60\ ^{\circ}$C. Additionally, it possesses a response time of less than 5 µs. These sensors enable our Robo Twin's hand to accurately detect and measure applied forces.

Although the FSR is a nonlinear device that is highly sensitive to changes at low forces and much less sensitive to changes at high forces, a voltage divider with nonlinear transfer characteristics is employed to provide greater values of output at smaller values of sensor resistance. In most cases, a resistor with a value of 3.3 kΩ is considered good, as this value lies within the range of the FSR's resistive output.

However, it is recommended that the sensitivity of FSRs be increased to a higher value at lower forces, and at higher forces, a lower value of the static resistor must be used. Since our goal here is to measure forces applied while performing a handshake, which mostly fall in the lower force range, the choice of higher static resistance serves the purpose of the study. Therefore, 10 KΩ was used as a static resistor to form a voltage divider circuit with the FSR.

The FSR sensor was calibrated up to a force value of 10 N by changing the known applied force on the FSR's sensing area. The data accumulated during the calibration process are plotted to visualize the relationship between the applied force, measured resistance, and calculated conductance of the FSR sensor. It is clear from Figure 7a,b that the FSR's conductance is good, as it varies almost linearly with the applied force, unlike the FSR's measured resistance, which exhibits a nonlinear relationship.

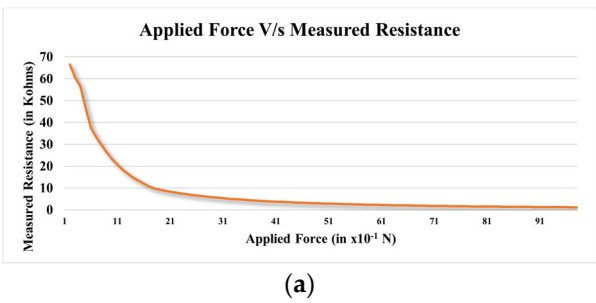
**(a)**

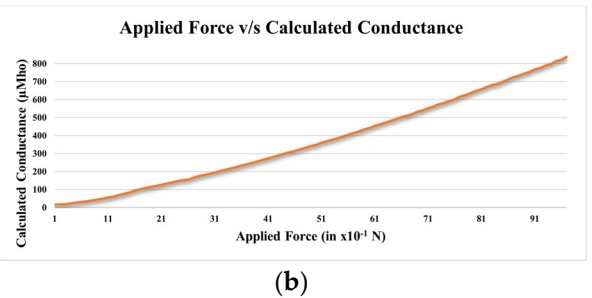
**(b)**

**Figure 7.** (**a**) Graphical relationship between applied force and measured resistance; (**b**) graphical relationship between applied force and calculated conductance.

To achieve more detailed insight into the obtained data, the correlation coefficient, a measure of similarity between the two sets of data, was calculated. The obtained correlation coefficient is tabulated in Table 1. The dataset of applied force and conductance showed the highest similarity, with 'r' being almost equal to 1. This indicates that the conductance parameter has a strong positive correlation with the applied force. Hence, to predict the applied force on the force-resistive sensor, the value of the calculated conductance was used rather than the direct measured resistive output.

**Table 1.** Correlation coefficient 'r' for different datasets.

| Dataset | Correlation Coefficient 'r' |
| --- | --- |
| Applied force and FSR's output resistance | $-0.5179$ |
| Applied force and FSR's output conductance | 0.9906 |

A simple linear regression that uses the least-squares method to find the best relationship for a set of paired data was then used to estimate the line of best fit. In our case, this regression equation can estimate the value of applied force (as a dependent variable Y) from the calculated output conductance value (as an independent variable X) and is

given by the equation ŷ = mX + c. Running the linear regression test on the obtained paired dataset, we obtained the following equation:

$$\text{Applied Force (in N)} = 0.01 \times \text{Conductance (in μMoh)} - 0.83 \tag{1}$$

The line of regression represented by the above equation is illustrated in Figure 8, as shown below.

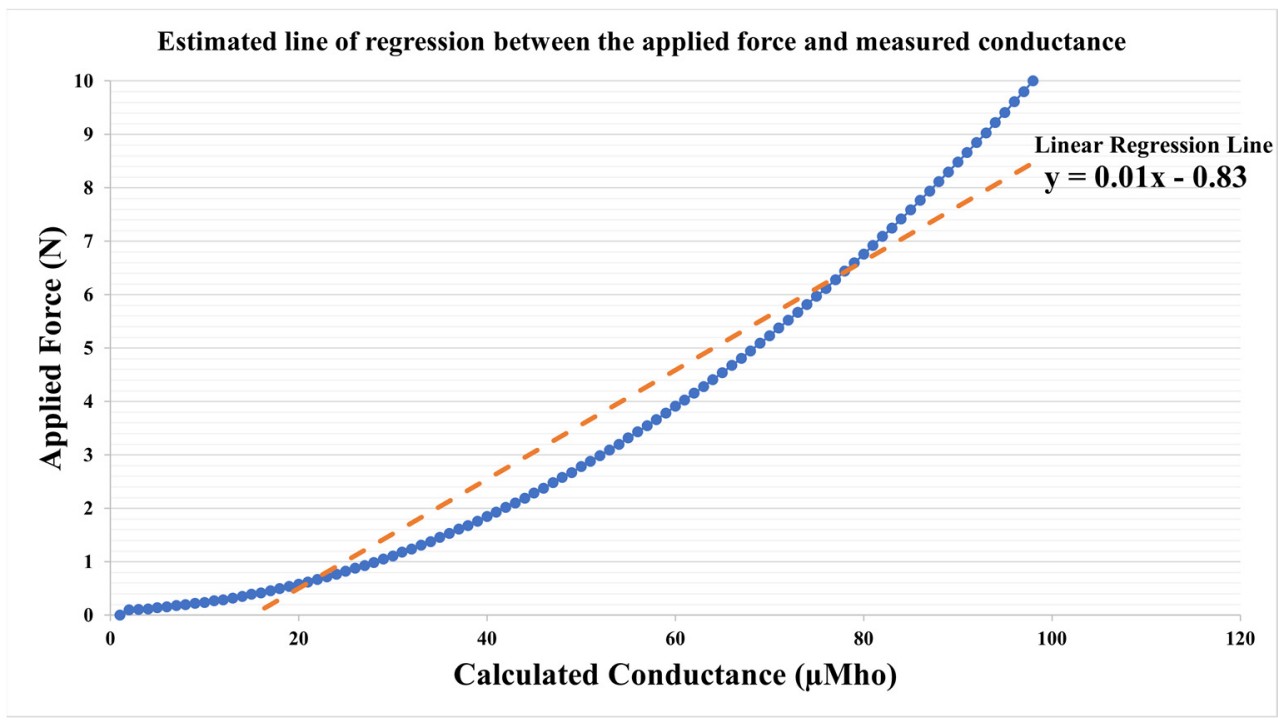

**Figure 8.** Estimated line of regression between the applied force and measured conductance.

As seen from the above regression line graph, the predicted force deviated from the actual values in the initial, middle and final stages. To overcome these deviations and to make our prediction more accurate, we decided to divide the conductance into four different ranges, i.e., from (0 to 120) μMho; (121 to 250) μMho; (251 to 600) μMho; and (>600) μMho. The estimated line of regression for each of these ranges is tabulated in Table 2.

**Table 2.** Estimated line of regression for different ranges of output conductance.

| Range | Estimated Line of Regression |
|---|---|
| (0 to 120) μMho | Applied Force (in N) = 0.00473 × Conductance (in μMoh) + 0.0347 |
| (121 to 250) μMho | Applied Force (in N) = 0.00889 × Conductance (in μMoh) − 0.4670 |
| (251 to 600) μMho | Applied Force (in N) = 0.01273 × Conductance (in μMoh) − 1.5289 |
| (>600) μMho | Applied Force (in N) = 0.01613 × Conductance (in μMoh) − 3.4949 |

Finally, to validate the calibration of the FSR sensor, a few independent tests were run by placing known standard masses of 0 g, 50 g and 100 g on the FSR's sensing area. The results obtained from these validation tests are tabulated in Table 3. The calibration results show the measurement accuracy of FSR sensors was approximately 98%.

A total of six FSR sensors, five Sparksfuns with a 0.5" sensing diameter and one Teck scan with a 1" sensing diameter were used to equip the force-sensing regions R1, R2 and R3, as presented in Figure 9.

**Table 3.** Calibration results of FSR.

| Standard Weight (grams) | Equivalent Force (N) | Calibration Result (N) |
| --- | --- | --- |
| 0 | 0 | 0 |
| 50 | 0.49 | 0.50 |
| 100 | 0.98 | 0.96 |

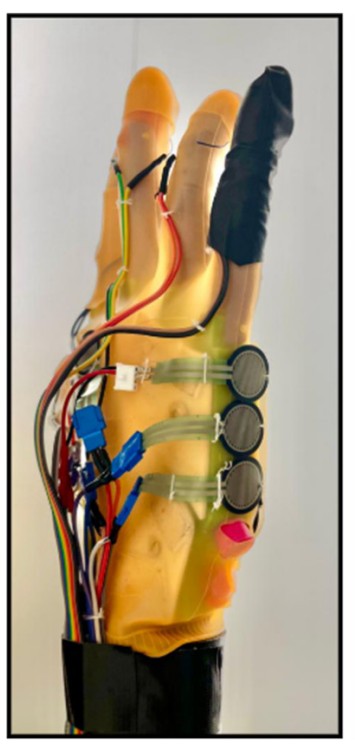
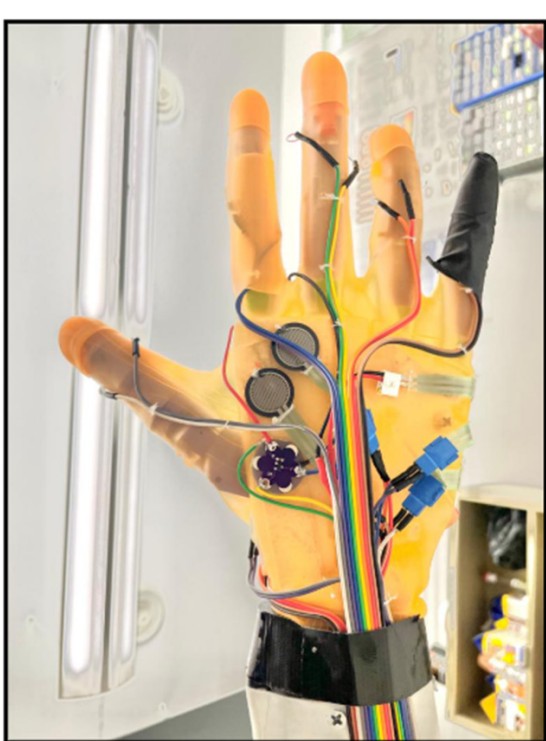
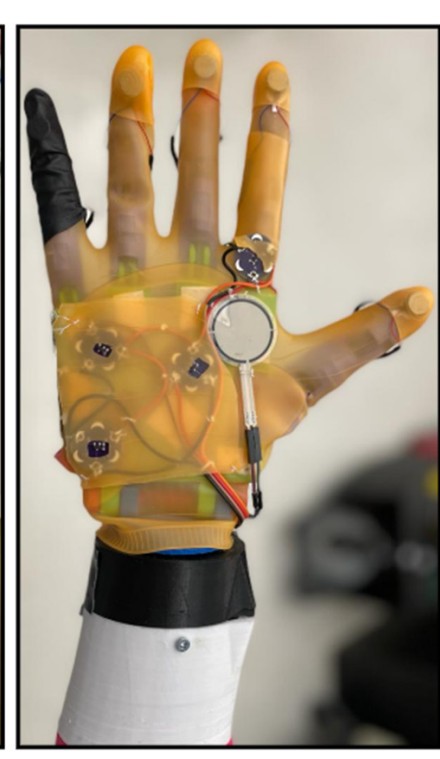

Placement of 3 Sparksfuns' 0.5" sensing dia FSR's in Region 1

Placement of 2 Sparksfuns' 0.5" sensing dia FSR's in Region 2

Placement of 1 Teckscans' 1" sensing dia FSR's in Region 3

**Figure 9.** Placement of six force sensors on an actual Robo Twin Hand.

### 5.2. Temperature Sensors

Sensing the temperature of the Real Twin's hand while performing a handshake is another important task of our Robo Twin's hand. Thermistor is the most widely used temperature sensor for the measurement of the temperature of objects in contact with robotic hands [7]. Therefore, we incorporated the Lilypad MCP9700 temperature sensor into our design for temperature sensing. The Lilypad MCP9700 is a small, low-powered linear active thermistor-type sensor. It offers an operating temperature range from −40 °C to +150 °C and provides ±2 °C accuracy. With a typical low operating current of 6 μA, this sensor ensures efficient and reliable temperature measurements in our system. It not only provides ease of calibration and implementation but can also be sewn into any fabric or material, such as silicon in our case. Moreover, these sensors are even washable, so there is no fear of damage to sensors due to any sweat encountered while performing a handshake.

From the information made available through the dataset of this temperature sensor, the voltage output by the sensor is linearly proportional to the Celsius temperature, i.e., 10 mV for every degree rise in temperature with a 0.5 V level set for 0 °C. It has a thermal response time of 1.3 s. Therefore, once we know the output voltage of the sensor, we can

calculate the temperature of the object in contact with this temperature sensor using the following equation:

$$\text{Temperature in Celsius } (^\circ C) = (\text{Sensors output voltage } - 0.5) \times 100 \tag{2}$$

Due to design constraints, for the placement of the temperature sensors on the Robo Twin hand, we only focused on sensing the inside temperature of the Real Twin's hand, i.e., the palm and the thumb. A total of four Lilypad MCP9700 sensors were used to sense the temperature of the inside palm and one for the thumb, covering the temperature sensing regions R1 and R2. The placement of the temperature sensors on the actual Robo Twin hand is demonstrated in Figure 10. A small cut in the silicone glove was made just over the temperature sensor to prevent the silicone from interfering with the actual temperature reading.

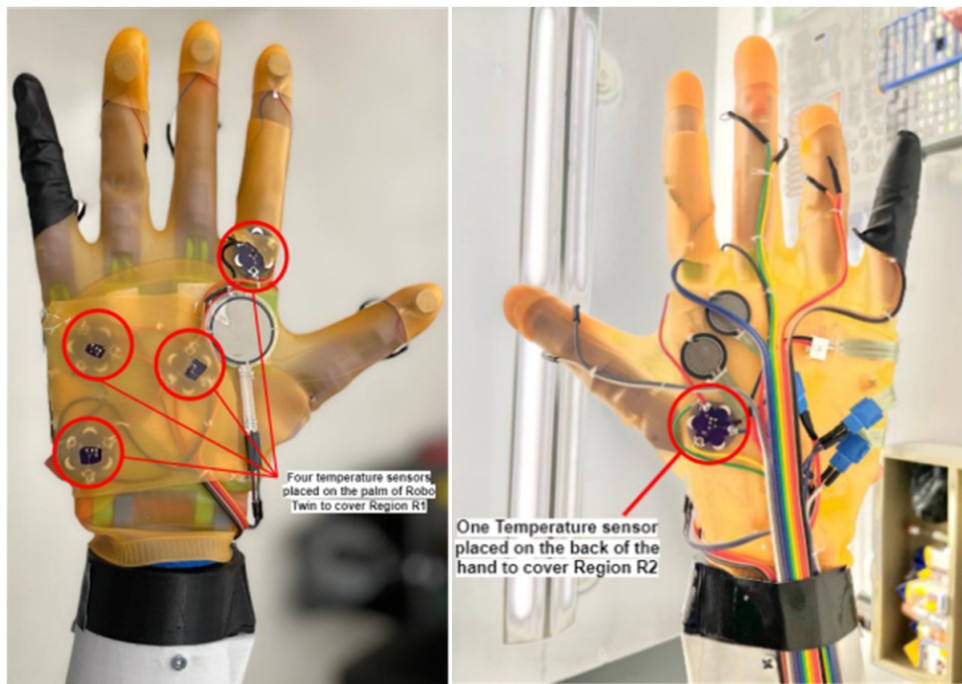

**Figure 10.** The actual placement of temperature sensors to cover sensing regions R1 and R2.

### 5.3. Vibrotactile Force Actuators

To provide haptic feedback on either side, so that each Real Twin can experience and feel the equivalent amount of force being applied by the other Real Twin, we decided to use eccentric rotating mass (ERM)-type vibration motors rated at 13,000 RPM (rotations per minute) and operating at 5 V DC as vibrotactile force actuators in our Robo Twin's hand design.

To make the haptic feedback more effective and closely imitate the force applied by the Real Twin's hand on the Robo Twin's hand, a Sparksfun DRV2605L haptic motor driver was used to operate each of these vibrotactile ERM motors. This haptic motor driver provides various inbuilt functionalities. However, we are more interested in the PWM (pulse width modulation) input (0% to 100%) for the duty-cycle control range functionality. This inbuilt functionality allows us to control the frequency of the vibrations of the motors based on the received input PWM (pulse width modulation) signal from the microcontroller, therefore providing the equivalent gradual effect of the applied force sensed by the corresponding FSR sensors.

We were only able to implement the feedback on regions R1 and R2 of the Real Twin's hand, not region R3, as illustrated in Figure 5b, due to design constraints. The actual placement of these actuators covering regions R1 and R2 is shown in Figure 11.

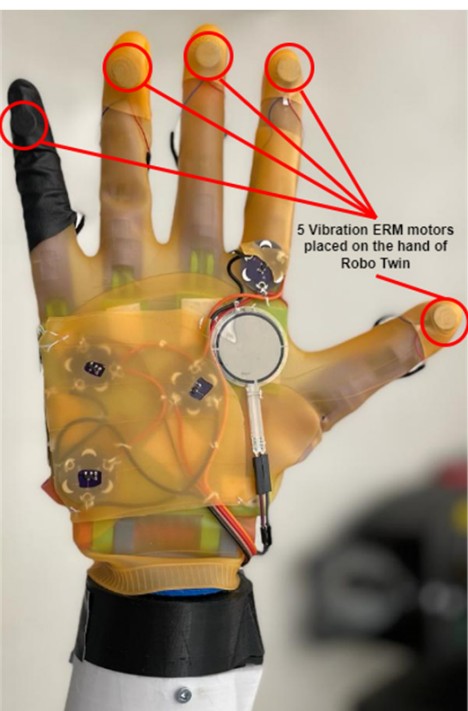

**Figure 11.** Placement of vibration ERM motors on actual Robo Twin hand for haptic feedback.

## 6. Performance Evaluation

### 6.1. Experimental Objective

The overall experience of haptics and human–robot interaction is highly dependent on one's personal preferences. These are more apparent in a situation such as a handshake, as every person has their own personality. Through our designed Robo Twin hand and this experiment, we aimed to enhance our understanding of the participants' handshake characteristics, such as their grasp, hand temperature and the force applied by various regions of the hand during a handshake, and to find a correlation between these handshaking characteristics and the gender and personality traits of the participants. We also aimed to evaluate the overall human–robot interaction as perceived by the participants while performing a handshake and to assess how open the users were to our Robo Twin concept.

### 6.2. Experimental Setup

To evaluate the performance of this designed system, two right Robo Twin hands were mounted on a 3 ft pole on either side. Each hand was connected to a laptop via a USB A to B converter cable to record the experimental data. A chair was kept in front of the hands to allow the participants to sit comfortably and perform a handshake with the Robo Twin hands.

A total of 18 participants consisting of 8 male and 10 female subjects took part in both the quantitative and the qualitative analysis of the handshake experience with the Robo Twin. For each experiment, two participants were required on each side of the system. Prior to the experiment, the researchers explained to the participants how they should interact with the Robo Twin arm, and a detailed step-by-step explanation of the experiment was provided. At the end of the handshake experiment, participants were asked to complete a short questionnaire to provide their overall feedback on their interaction and experience with our system. Additionally, they were invited to take the BIG FIVE personality assessment on the ITP metric platform. This test assesses one's five personality traits, namely extraversion, emotionality, conscientiousness, agreeableness and openness [40].

*6.3. Experimental Results*

The experimental findings are classified into qualitative and quantitative analysis, where the qualitative analysis reflects the overall haptics and human–robot experience as perceived by the participants and the quantitative analysis aims to provide insight into physical handshaking characteristics and their correlation to the gender and the personality traits of the participants.

6.3.1. Qualitative Analysis

The qualitative analysis of our system was achieved through a post-experiment questionnaire with a total of 12 questions based on different experiences that the participants had during their handshake with the Robo Twin arm. Moreover, these questions used the Likert scale, where the participants were asked to rate their experience on a scale of '1' to '5', with '1' indicating the lowest rating and '5' indicating the highest rating, rather than just a 'Yes' or 'No'. It is interesting to note that 61.1% of participants said that they would use our system to connect with their loved ones remotely and perform a handshake, and 27.8% of them had a neutral perspective about our system.

6.3.2. Quantitative Analysis

Our Robo Twin hand design helped us to collect data and to study the various characteristics of a human handshake, such as the average duration of a handshake, average and maximum force applied by the participants over the period of their handshake and the average temperature of the participants' palm and thumb:

- **Duration of handshake (in seconds):** The participants were told to begin a handshake with a standard grasp, comparable to how they would normally interact with others daily, and then progressively tighten the grip. This deliberate fluctuation in grip strength allowed users to observe how the robotic system gradually changed the amount of force it applied in response to the force that was perceived during the handshake. The length of the handshake that was recorded corresponds to the beginning and end of the specified handshake. On an average, the duration of the handshake for females was '7.88 s', whereas for male participants, it was '11.03 s', being approximately '3 s' longer than that of the female participants. The results obtained are in accordance with the study [41], which found that males tended to have a longer duration of handshakes than their female counterparts. The average duration of handshakes for both males and females is illustrated in Figure 12a.

- **Average force applied (in Newtons):** The average force applied by each participant while performing a handshake is the average value sensed by all force sensors placed on the Robo Twin hand over the duration of the participant's handshake. By averaging the values of force applied by individual male and female participants, we found that males tended to apply comparatively more force than their female counterparts, in conformity with [41,42]. This difference in the average force applied by each gender during the duration of the handshake is illustrated in Figure 12b.

- **Maximum force applied (in Newtons):** The maximum applied force is the maximum value of force sensed by the FSR sensors placed on the hand during the entire course of the handshake by the participant. We found that the maximum force that a male candidate applied during the handshake with the Robo Twin hand was 10.08 N, while the maximum force that a female candidate applied was 5.97 N. Calculating the average of maximum applied force by gender, this average was 4.58 N for females, whereas it was 7.11 N for males. Hence, during the handshake, male participants tended to apply significantly more force than the female participants, as found in [42]. Additionally, we observed that some of the male participants applied less force than some of the female participants. This is in accordance with the study [43], which measures the hand grip strength in young men, women and highly trained female athletes using a handheld hand grip ergometer instead of human-provided ratings, as in [41]. This study states that 5% of female participants applied more force than 10%

of male participants. Therefore, not all males will have higher values of applied force than females, and not all females will have lesser values of applied force than males.

- **Palm and thumb temperature (in °C):** With the help of temperature sensors integrated on the Robo Twin hand, we were able to sense and record the average temperature of the participants' palm and thumb for the duration of a handshake. We found that for both the average temperature of the palm or the thumb, the male participants' temperature was maintained at a slightly higher value than the female participants' temperature. Additionally, when comparing the temperature of the palm with the thumb, whether it is male or female participants, we noticed that the palms of the participants' hands were maintained at a higher temperature than the thumbs. These details are illustrated in Figure 13.

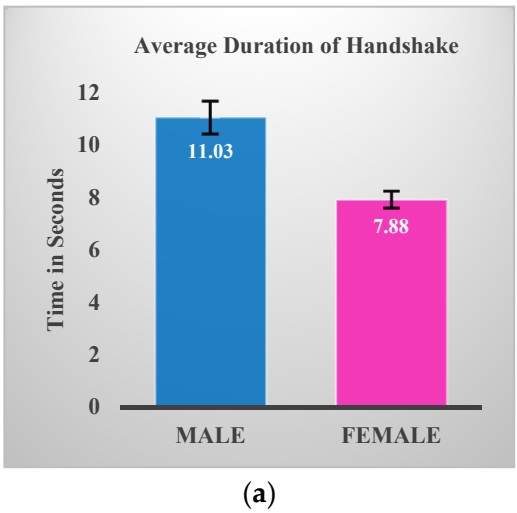
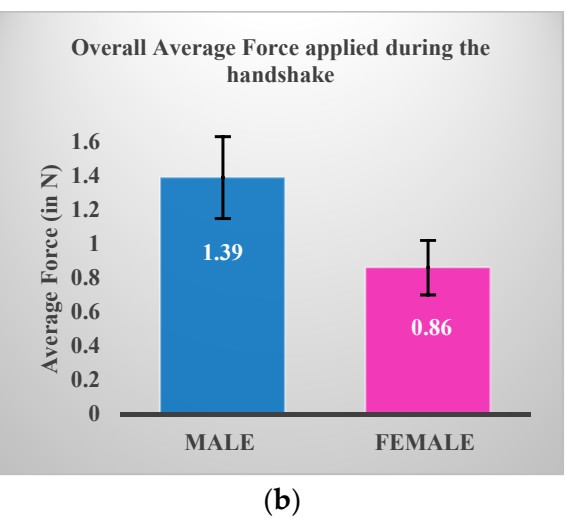

(**a**)   (**b**)

**Figure 12.** (**a**) Average duration of handshake by gender; (**b**) average force applied during handshake by males and females.

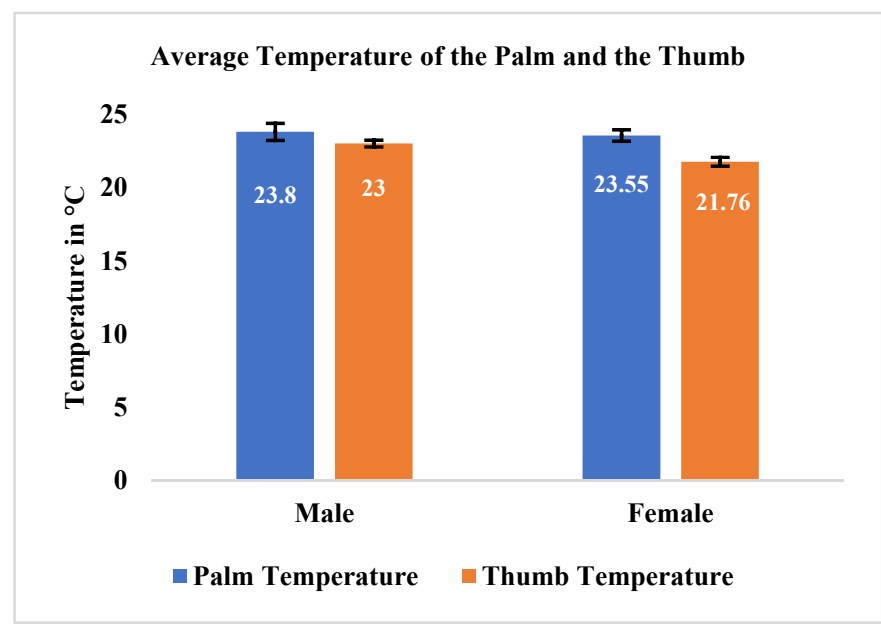

**Figure 13.** Average palm and thumb temperature in male and female participants.

- **Personality and handshaking characteristics:** The participants' personality traits were assessed through a report generated by the ITP (Individual and Team Performance) metric platform based on the responses to the questionnaires that the partici-

pants completed. This report outlines the level of each participant's five personality factors, namely extraversion, emotionality, conscientiousness, agreeableness, and openness. The score for each trait is presented in percentile form, with categories of 'Low' (between 0th and 25th percentile), 'Moderate' (between 25th and 75th percentile) and finally 'High' (more than 75th percentile). For the male participants, we found that those who applied the least force and had a short duration while performing a hand-shake scored high on conscientiousness and emotionality but low on the extroversion, agreeableness, and openness scales. Meanwhile, participants who applied the most force and had a longer duration scored high on conscientiousness, agreeableness, and openness, but moderately on the extraversion and emotionality scales. We represent these results with a four-quadrant plot with 'Duration' on the *X*-axis and 'Average applied force' on the *Y*-axis in Figure 14.

For the female participants, we found that the participants with firmer handshakes were more open to the experience only if they scored high on conscientiousness, emotionality, agreeableness and low to moderately on the extraversion scale of personality traits. Meanwhile, the participants with less-firm handshakes were found to be less open to the experience, provided they scored low on conscientiousness and agreeableness, moderately on emotionality and low to moderately on the extraversion scale of personality traits. The results obtained are generally in conformity with [41,42], which showed that the individuals with longer and firmer handshakes were more open to the experience. However, we believe that to be able to comment more on the correlation between personality traits and handshake characteristics, a larger dataset is needed.

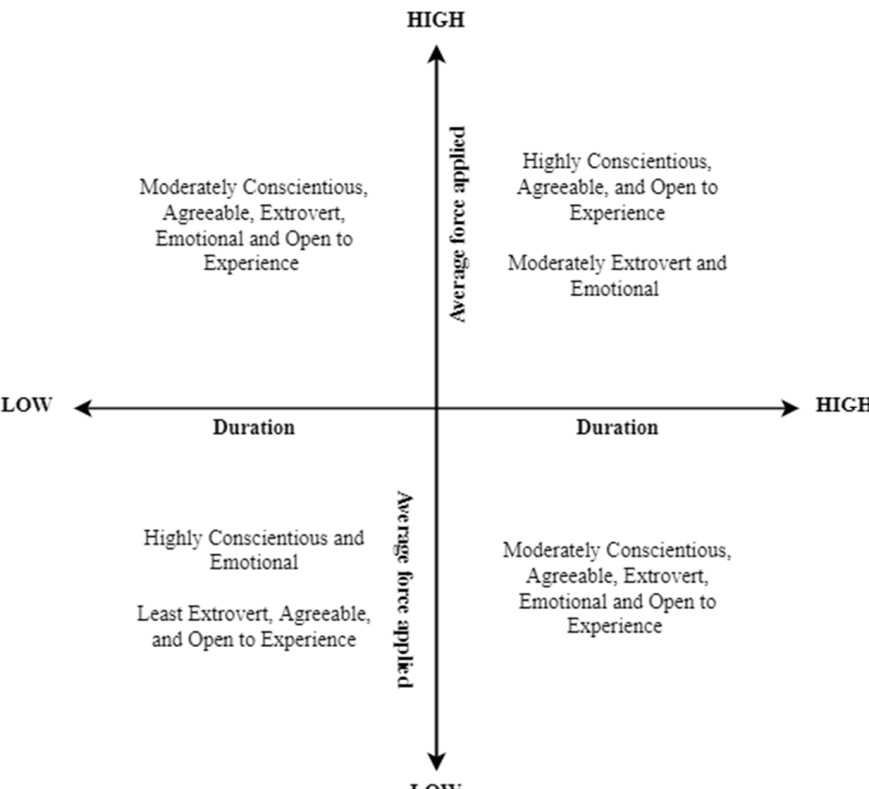

**Figure 14.** Four-quadrant plot of personality traits for male participants across duration and average applied force.

Although the implementation of the Robo Twin arm was primarily focused on replicating a human handshake here, it can be utilized in various other applications. For instance, it can mimic different types of greetings and gestures, such as high fives, fist bumps or hugs, thereby increasing the range of virtual haptic interactions. This enables users to engage

in more immersive emotional expression and connection building. The preservation and study of cultural practices and traditions can be facilitated using Robo Twin arms. Users may virtually shake hands with people from other civilizations, fostering respect for and knowledge of other cultures.

## 7. Limitations

Although our results are promising, there are some limitations to this study: firstly, the relatively small dataset (i.e., only 18 participants); secondly, for some participants with relatively small hand sizes (mostly females), their grasp did not reach all the sensors during the handshake, especially the sensors located in regions R1 and R2 of the Robo Twin hand, impacting the calculation of the average and maximum force applied for these participants. We believe the design can be improved to accommodate all sizes of hands and to overcome the design constraints, and the dataset can be increased for more precise insight into the findings.

## 8. Conclusions

This paper puts forwards the design and implementation of a Robo Twin hand integrated with temperature and force sensors and vibrotactile force actuators. This Robo Twin hand is designed to imitate the human handshake. Moreover, it allows two physically distant individuals to perform a handshake. After analyzing the system's performance both qualitatively and quantitatively, we propose the following future improvements, which could make the Robo Twin hand and system in general more intuitive.

From the perspective of Robo Twin hand design, it needs to be modified to accommodate both male and female hand sizes, as a few of our female participants highlighted that the Robo Twin hand was too bulky to allow them to completely grasp it as they would normally do while performing a handshake with any other person. In order to enhance the arm's performance, fidelity and naturalness in handshake exchanges and to provide users a more realistic and immersive handshake experience, we will incorporate actuation mechanisms for both the bicep and the shoulder.

Furthermore, since our main objective in designing the Robo Twin hand system is to allow people to perform a handshake remotely, we need to establish a bidirectional communication network that would allow the transmission and reception of such sensory data to achieve a virtual handshake remotely on the other side of the system. Additionally, both temperature and force are very important sensory and motor functions of the human hand; therefore, the advanced version of the Robo Twin hand will have to have temperature sensors for sensing all the regions of the Real Twin's hand that are in contact with it while performing a handshake, instead of just sensing the palm's and the thumb's temperature. Moreover, it needs temperature actuators to actuate the sensed temperature on the other hand, such as the force, to improve the user's experience and feeling of a handshake with our system. Another potential use could be to create immersive experiences by integrating the Robo Twin arms with VR or AR technology, providing users with an engaging and realistic interaction that goes beyond a simple handshake. As a result, users will have a greater feeling of presence and immersion, giving them the impression that they are actually engaging with other people in a shared virtual area. For example, users can shake hands with virtual people or objects in simulated environments, enhancing the interactivity of the simulations. These underactuated robotic arms can also find applications in physical therapy and rehabilitation settings to assist patients remotely with hand exercises and rehabilitation activities to create a more engaging and tailored experience. Furthermore, robotic arms can track patient motions and progress, allowing for the monitoring and modification of treatment schedules.

Given the preliminary nature of the research and the resources available, we chose a small sample size to acquire preliminary insights into the viability and potential of the Robo Twin arm. It is crucial to note, nevertheless, that a bigger sample size would have allowed for a more thorough analysis of the impacts and consequences of this technology. In future

work, we intend to widen the participant pool to incorporate a more varied spectrum of people and increase the sample size to guarantee a more thorough analysis. This will allow us to resolve any potential sample size issues and improve the validity of our conclusions.

Therefore, a lot of potential research work can be carried out to propose ideas and solutions for robotic hands with haptic modalities, especially in replicating a human handshake. Further research is warranted to answer some of these questions and provide more insights into the design and development of DT robotic representation.

**Author Contributions:** Conceptualization, A.E.S.; Methodology, A.E.S., F.L. and M.F.; Software, M.F.; Validation, F.L. and A.E.S.; Investigation, M.F. and F.L.; Writing—original draft, M.F.; Writing—review & editing, A.E.S. and F.L.; Supervision, A.E.S. All authors have read and agreed to the published version of the manuscript.

**Funding:** This research received no external funding.

**Institutional Review Board Statement:** This study, which involved human participants, was conducted in accordance with the Tri-Council Policy statement (2014) and other applicable laws and regulations and approved by the Research Ethics Board (or Ethics Committee) of University of Ottawa (Ethics File Number: H-01-23-8720 approved on 7 February 2023).

**Informed Consent Statement:** All participants gave their informed consent for inclusion before they participated in the study.

**Data Availability Statement:** Due to privacy concerns and in accordance with the ethics approval obtained for this research study, the data collected and analyzed in this paper is not publicly available.

**Conflicts of Interest:** The authors declare no conflict of interest.

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
