# Peer review of "Digital Twin Haptic Robotic Arms: Towards Handshakes in the Metaverse"

_electronics, doi:10.3390/electronics12122603_

Round 1

Reviewer 1 Report

This research paper presents the development and execution of a Robo Twin hand that incorporates temperature and force sensors, along with vibrotactile force actuators. The primary objective of this Robo Twin hand is to replicate the experience of a human handshake. Additionally, it enables individuals who are physically separated to engage in a handshake gesture. The verification seems clear and quite enough. The idea sounds interesting.  Please add more analysis about the usage for this application. A support video is also welcomed.

Reviewer 2 Report

The article appears to be designing a robotic arm for use in interacting within the metaverse to enhance user experience. My decision is not to recommend this article, and below is my detailed feedback:

(1) Designing a robotic arm specifically for handshakes seems unnecessary and wasteful, as there are already devices available that enhance user experience and are not limited to handshakes.

(2) In the metaverse, users not only desire a good handshake experience, but also need to be able to perceive more of the environment to obtain a richer experience.

(3) The suggestion I provided is to try using more precise materials to construct the force-conducting device, and to study different movements. This would not only easily achieve the handshake experience but also allow for the completion of other activities.

Reviewer 3 Report

The following is my comments.

(1) The text in Fig.2 is too small. 

(2) The printing material and technology details should be shown.

(3) Space should be inserted at the begining of the paragraph.

(4) The manufacturer, sources, model and key parameters should be given in details for the sensors, electrical devices, parts and instruments. 

(5) The text in Fig.10 is too small.

(6) What is the SNR or other parameters that can be used to evaluate your instruments? 

Reviewer 4 Report

Dear Researcher,

tank you for your work.

As I see, the aim of the research is a crossline between Engineering and Psycological aspects, and it’s very intresting mixture.

However there are some aspects that should be clarified

* In handshacke, the bicipets actuation is very importante. In Figure 3, is clearly visibile a couple of rotational actuators that seems to simulated this motion. But for your experiment phase, ther’s no picture that cleary show the interation between human and robotic harm, especially for bicipets actuated vertical motion. Please, attach more picture of esperimental phase

2  * Right for psychological sound of research, I think is important to improve it for experimental cluster, with muliples subject of differente cultures and backgrounds: the somatic aspect of the handshake is very different according to this factor

3  *    For the realization of robotic harm, you declare to use a PVC material 3d Printed with ultimaker 2+. To the best of my personal knowledge, ther is no PVC material printable...at least, not with ultimaker 2. Please, show evidence of the used material for 3d Printing.

4 * You declare a nozzle size of 6mm. Just like for previous point, please show evidence of this aspect. My personal experience on 3D printing is not compatible with this nozzle size hypothesys for this 3d printer machine.

5 *  A population of 18 participants is too small. Please, improve the experiment with a consistent population (since this is a procedure that does not require any type of invasiveness, I suggest at least 50 participants of different cultures and/or ethnic groups and/or geographical origins)

6 *   On raw 405, you talk about an average handshacking time of 11.03 sec. In my opinion, and for my experience, is a very long time. Please, consider the possibility to validate this timing improving licterature research.

Round 2

Reviewer 2 Report

Can be accepted.

Reviewer 4 Report

Dear Researchers,

thank you for your reply and  your article revisions.

Best Regards